# Sodium Thiosulphate-Loaded Liposomes Control Hydrogen Sulphide Release and Retain Its Biological Properties in Hypoxia-like Environment

**DOI:** 10.3390/antiox11112092

**Published:** 2022-10-24

**Authors:** Lissette Sanchez-Aranguren, Milda Grubliauskiene, Hala Shokr, Pavanjeeth Balakrishnan, Keqing Wang, Shakil Ahmad, Mandeep Kaur Marwah

**Affiliations:** 1Aston Medical School, College of Health and Life Sciences, Aston University, Birmingham B4 7ET, UK; 2Mirzyme Therapeutics, Innovation Birmingham Campus, Faraday Wharf, Holt Street, Birmingham B7 4BB, UK; 3Pharmacy Division, School of Health Sciences, Faculty of Biology, Medicine and Health, University of Manchester, Manchester M13 9PL, UK; 4School of Engineering and Technology, College of Engineering and Physical Sciences, Aston University, Birmingham B4 7ET, UK

**Keywords:** liposomes, controlled-release, drug delivery systems, hydrogen sulphide, angiogenesis, mitochondrial metabolism

## Abstract

Hypoxia, or insufficient oxygen availability is a common feature in the development of a myriad of cardiovascular-related conditions including ischemic disease. Hydrogen sulphide (H_2_S) donors, such as sodium thiosulphate (STS), are known for their cardioprotective properties. However, H_2_S due to its gaseous nature, is released and cleared rapidly, limiting its potential translation to clinical settings. For the first time, we developed and characterised liposome formulations encapsulating STS and explored their potential for modulating STS uptake, H_2_S release and the ability to retain pro-angiogenic and biological signals in a hypoxia-like environment mirroring oxygen insufficiency in vitro. Liposomes were prepared by varying lipid ratios and characterised for size, polydispersity and charge. STS liposomal encapsulation was confirmed by HPLC-UV detection and STS uptake and H_2_S release was assessed in vitro. To mimic hypoxia, cobalt chloride (CoCl_2_) was administered in conjunction with formulated and non-formulated STS, to explore pro-angiogenic and metabolic signals. Optimised liposomal formulation observed a liposome diameter of 146.42 ± 7.34 nm, a polydispersity of 0.22 ± 0.19, and charge of 3.02 ± 1.44 mV, resulting in 25% STS encapsulation. Maximum STS uptake (76.96 ± 3.08%) from liposome encapsulated STS was determined at 24 h. Co-exposure with CoCl_2_ and liposome encapsulated STS resulted in increased vascular endothelial growth factor mRNA as well as protein expression, enhanced wound closure and increased capillary-like formation. Finally, liposomal STS reversed metabolic switch induced by hypoxia by enhancing mitochondrial bioenergetics. These novel findings provide evidence of a feasible controlled-delivery system for STS, thus H_2_S, using liposome-based nanoparticles. Likewise, data suggests that in scenarios of hypoxia, liposomal STS is a good therapeutic candidate to sustain pro-angiogenic signals and retain metabolic functions that might be impaired by limited oxygen and nutrient availability.

## 1. Introduction

Cardiovascular diseases represent a major global health problem and are a leading cause of morbidity and mortality worldwide according to the World Health Organisation [1]. Conditions such as ischemic heart disease and cardiomyopathies have a common pathophysiological mechanism linked to insufficient oxygen and nutrient supply to the cardiac muscles [2]. This vascular supply insufficiency usually occurs when the arterial oxygen delivery is below vascular tissue demand [3], leading to tissue ischemia and impaired angiogenesis. Angiogenesis, or the generation of new blood vessels, is hallmarked by the upregulation of hypoxia inducible factor-1 α (HIF-1α) which triggers hypoxia-inducible genes, such as vascular endothelial growth factor (VEGF) [4]. Recently, therapeutic angiogenesis has gained increased recognition in ischemic disease especially as cardioprotective therapies such as β-blockers often fail in preventing the development of a range of ischemic cardiovascular conditions such as acute coronary syndrome and angina [5]. This signifies the ongoing need for effective therapeutic approaches tackling defective molecular mechanisms for this patient cohort.

Hydrogen sulphide (H_2_S) is a gaseous molecule with the characteristic foul odour of rotten eggs that was previously considered to be poisonous [6]. However, research has demonstrated that mammalian cells produce low levels of H_2_S as an important signalling molecule [7,8,9]. It has multiple effects in many body organ systems including the central and peripheral nervous systems, cardiovascular, gastrointestinal and respiratory systems [6] with the ability to regulate cellular metabolism [9,10], angiogenesis [11], inflammatory responses [12] and oxidative stress [13] through several mechanisms including the preservation of the mitochondrial metabolic and cellular functions [9,14]. Exogenous H_2_S has been shown to be cardioprotective in various experimental models of cardiac injury [15,16], offering a potential therapeutic option where a lack of its endogenous availability has resulted in myocardial ischemia and reperfusion injuries [17,18,19]. Sodium thiosulphate (Na_2_S_2_O_3_, STS), a H_2_S donating sulphur salt [20,21,22], is an industrial compound with a long history of medical use specifically in the treatment of cyanide poisoning [23,24,25]. STS has been already explored in the treatment of acute coronary syndrome observing a therapeutic benefit in the preservation of cardiac function after myocardial ischemia [26]. 

Irrespective of the strong clinical importance of H_2_S, the application of therapeutic H_2_S donors is limited in effectiveness due to poor biodistribution, rapid release of H_2_S, clearance and lack of selectivity [27]. These challenges may be addressed by developing controlled release drug delivery systems in order to delay donor, thus H_2_S, degradation and clearance whilst retaining biological actions. Nanoparticles such as liposomes have gained increased interest in drug delivery research as they are safe to administer in humans [28,29] and may provide improved therapeutic effects by stabilizing encapsulated reagents, increasing cellular and tissue uptake, and improving biodistribution of therapeutic reagents to target sites in vivo [30]. 

The objective of this study is to investigate and characterise liposomes entrapping STS and to explore the effect of formulation parameters on liposome characteristics, STS and H_2_S release. Furthermore, we assessed the ability of liposome encapsulated STS to retain biological properties of H_2_S in endothelial cells exposed to a hypoxic-like environment, using cobalt chloride (CoCl_2_). 

## 2. Materials and Methods

### 2.1. Reagents and Drugs

Soy phosphatidylcholine (PC) was obtained from Avanti Polar Lipids (Avanti, Tonawanda, NY, USA, grade, ≥99%). Sodium thiophosphate, tetrabutylammonium hydrogen sulphate, methanol, potassium dihydrogen phosphate, octaethylene glycol monododecyl ether, acetic acid and 3-(4,5-dimethyl-2-thiazolyl)-2,5-diphenyl-2H-tetrazolium bromide (MTT) and polycarbonate filters (pore size 400 nm, 200 nm and 100 nm, WHA800309) were all obtained from Sigma-Aldrich (Dorset, UK). 1,1′-Di-n-octadecyl-3,3,3′,3′-tetramethylindocarbocyanine perchlorate, 97% (DilC) was obtained from Alfa Aesar (Lancashire, UK). Ultrapure water was obtained from a Milli-Q purification system (Millipore, Billerica, MA, USA). 

### 2.2. Preparation of STS Loaded Liposomes

Liposomes were prepared by the ethanol injection method established by Batzri and Korn, 1973 [31]. Lipids were first dissolved in ethanol. The optimal lipid composition for encapsulation STS was investigated by varying the ratio of cholesterol and anchor lipids PC and DOTAP as shown in Table 1. The total lipid concentration in the final formulation (lipids and cholesterol together) was 10.0 mg/mL.

This was injected into 1 mL of 25 mg/mL STS in PBS buffer at a temperature above the transition temperature of the lipids. Liposomes were then extruded through polycarbonate membranes using Avanti mini-extruder (Stratech Scientific Ltd., Cambridge, UK) at a temperature above the transition temperature of the lipids. In order to preserve the stability and produce evenly sized liposomes, the extrusion process was as follows: 800 nm × 8400 nm × 8200 nm × 8 and 100 nm × 8. Ethanol and unentrapped STS were removed via dialysis using Slide-A-Lyzer dialysis cassettes (12–14 kDa MWCO) against buffer solution [32]. 

### 2.3. Liposome Characterisation: Particle Size, Polydispersity and zeta Potential

The mean liposome size and polydispersity index (measurement of homogeneity of vesicle sizes) of the liposome nanoparticles were measured by dynamic light scattering (DLS) using a Zetaplus (Brookhaven Instruments, Holtsville, NY, USA) following dilution with distilled water (1:5 ratio) to ensure intensity adjustment. A small polydispersity value (<0.2) indicates a homogenous vesicle population and a larger polydispersity (>0.3) indicates heterogeneity [33]. The particle charge was quantified as zeta potential (ζ) following dilution with distilled water (1:5 ratio). Zeta potential was measured by photon correlation spectroscopy using a Zetaplus (Brookhaven Instruments, Holtsville, NY, USA). 

### 2.4. HPLC Methodology

The detection of STS was assessed using a reverse phase HPLC method adapted from Schulz et al. [34]. A Shimadzu LC-2030C Plus RoHS—Prominence-I separation module HPLC with UV detection was utilised at an operating wavelength of 210 nm. A Phenomenex HyperClone™ column (5 µm C18 4.6 × 150 mm column) was used with a 10 μL sample injected at 27 °C. The mobile phase consisted of 0.005 M (1.698 g/L) tetrabutylammonium hydrogen sulphate dissolved in a solution of methanol-phosphate buffer (15:85). The phosphate buffer was 10 mM (136.086 g/mol therefore, 1.36 g/L) potassium dihydrogen phosphate with a pH of 7. The flow rate was set at 1.0 mL/min with a 10 µL injection volume. Stock solutions and standard solutions of STS were prepared with PBS ranging from 0.001–15 mg/mL. A final calibration curve with an R^2^ of 0.9868 and a linear equation of y = (4 × 10^6^) was obtained.

### 2.5. Determination of Entrapment Efficiency

The entrapment efficiency of STS in liposomes was determined following comparison of STS pre and post dialysis. Liposomes were dissolved upon the addition of 0.6 mM of octaethylene glycol monododecyl ether (C12E8) in a 10:1 ratio and then analysed using HPLC-UV analysis. The percentage encapsulation efficiency of STS in liposomal formulations was calculated using Equation (1): (1)$$E=\frac{S_{d}}{S_{t}}\times 100\%\;$$
where *E* is the encapsulation efficiency (%), *S_d_* is the STS content post-dialysis (mg) and *S**_t_* is the total STS content pre-dialysis (mg).

### 2.6. Cell Culture

EA.hy926, an immortalised endothelial cell line derived from human umbilical vein endothelial cells (HUVEC) was used as an endothelial cell model [35]. EA.hy926 were cultured and routinely maintained in DMEM containing 10% FBS and 100 U/mL penicillin/streptomycin maintained at 37 °C and 5% CO_2_ atmosphere.

For experiments mirroring hypoxia condition, EA.hy926 were exposed to CoCl_2_ (400 μM) for 24 h to mimic hypoxia in vitro. CoCl_2_ is a well-establish compound that artificially induces hypoxia by blocking the degradation of hypoxia inducible factor-1 α (HIF-1α), a marker of angiogenesis [36]. To observe the potential of formulations to modulate endothelial function, EA.hy926 were co-exposed with CoCl_2_ and either non-formulated or liposome-formulated STS for 24 h.

### 2.7. Cellular Uptake of STS

Formulations 4, 5 and 6 were evaluated against non-encapsulated STS in their ability to be up-taken by EA.hy926 cells after 2, 4 and 24 h. Cells were seeded into 24 well plates at 2.0 × 10^4^ cells per well. Washed formulations were diluted (10-fold) with fresh media to give 0.75 mg/mL final STS concentration and 400 µL added to each well. Media was replaced after 2 h for all cells to ensure only STS released from the liposomes was detected. Subsequently, at 2, 4 and 24 h media was removed, and cells were lysed following the addition of 0.6 mM of octaethylene glycol monododecyl ether. The resultant cell lysate was centrifuged for 10 min at 16,000× *g* rpm and analysed using HPLC-UV detection for STS content. 

Furthermore, formulation 6 was selected to visualise cellular uptake by loading liposomes with fluorescent marker DilC as this formulation gave the most controlled release of STS. DilC Liposomes were formulated with the addition of 25 µg DilC during the lipid mixing stage. EA.hy926 were plated in coverslips at density of 5.0 × 10^5^ cells per coverslip and allowed to attach overnight. Following, media was removed and DilC loaded liposomes diluted with 10 parts of supplemented media (as detailed in Section 2.6) was added and cells and incubated for 24 h at 37 °C, 5% CO_2_ atmosphere. Subsequently, coverslips were washed and fixed with 4% *w*/*v* paraformaldehyde for 5 min at room temperature. Coverslips were then mounted onto glass slides with the addition of a DAPI-containing mounting media (SlowFade™ Diamond Antifade Mountant with DAPI; ThermoFisher Scientific, Loughborough, England, UK). Coverslips were subsequently analysed, and images recorded at 60× using a Nikon Eclipse Ti-E inverted microscope (Nikon Instruments Inc., Melville, NY, USA).

### 2.8. Determination of H_2_S Release 

Liposome encapsulated and non-encapsulated STS were diluted 1 in 10 using fresh media and added to EA.hy926 cells. As free H_2_S is a strong reducing agent, we used the tetrazolium dye 3-(4,5-dimethyl-2-thiazolyl)-2,5-diphenyl-2H-tetrazolium bromide (MTT, Sigma-Aldrich, Dorset, UK) to quantify H_2_S. When MTT is reduced it forms a purple colour, formazan, that can be measured by colorimetry and tabulated against a calibration curve. Media was replaced after 0.5 h for all cells. Briefly, EA.hy926 were plated in 96-well plates at 2.0 × 10^4^ cells/well and allowed to attach overnight. Non-encapsulated and encapsulated STS were applied in 1 in 10 dilution (100 μL) and media was replaced after 0.5 h and collected at 1, 2, 3, 4, 6 and 24 h. Next, 100 μL MTT (5 mg/mL) was added to the collected media and changes in absorbance were recorded on a plate reader at 570 nm. The reaction was carried out in a humidified incubator at 37 °C with 5% CO_2_ atmosphere to mirror the cell culture conditions and minimise evaporation. A H_2_S calibration curve was generated by preparing serial dilutions of freshly dissolved Na_2_S. H_2_S generation is shown as an hourly change in absorbance with respective H_2_S values.

### 2.9. Quantitative RT-PCR (qPCR)

To assess the potential of liposome formulations to enhance pro-angiogenic signals under hypoxia, relative expression of VEGF^165^ mRNA was measured using qPCR. Briefly, EA.hy926 were plated at a density of 1.0 × 10^6^ cells/dish in 100 mm petri dishes and cells allowed to attach overnight. CoCl_2_ (400 μM) was co-administered along with either non-encapsulated or liposome encapsulated STS, for 24 h. After treatments, cells were lysed, and mRNA extracted using RNeasy kit (Qiagen, Hilden, Germany) as per the manufacturer’s guidelines. RNA was converted to cDNA using the EvoScript Universal cDNA Master (Roche Life Sciences, Basel, Switzerland) following the manufacturers’ instructions. Gene expression of human VEGF^165^, thioredoxin, Glut-1 and YWHAZ (housekeeping) were quantified by real-time PCR on a Lightcycler 480 (Roche Life Sciences, Basel, Switzerland) using the LightCycler 480 SYBR Green I Master and its specific primers (Primers sequences: VEGF^165^ F: 5′-GCAGAATCATCACGAAGTGGTG-3′, VEGF^165^ R: 5′-CACACAGGATGGCTTGAAGATG-3′, YWHAZ F:5′-CCTGCATGAAGTCTGTAACTGAG-3′, YWHAZ R: 5′-GACCTACGGGCTCCTACAACA-3′). Glut-1 F: 5′-GCTCATCAACCGCAACGAG-3′, Glut-1 R: 5′-TCATGGGTCACGTCAGCTGT-3′, Thioredoxin F: 5′-GTGAAGCAGATCGAGAGCAAG-3′, Thioredoxin R: 5′-CGTGGCTGAGAAGTCAACTACTA-3′. The expression of YWHAZ was used to calculate the relative expression of VEGF^165^, Glut-1 and thioredoxin mRNA. RT-PCR was performed using the following running conditions: Pre-incubation (1 cycle), amplification (45 cycles), melting curve (1 cycle), and cooling (1 cycle). Relative gene expression was calculated using the 2^−ΔΔCT^.

### 2.10. ELISA

The quantification of human VEGF in cell supernatant was performed using the Human VEGF DuoSet ELISA (DY293B) from R&D Systems, Abingdon, UK and performed according to the manufacturer’s specifications. Briefly, EA.hy926 were plated in 96-well plates at cell density of 2.0 × 10^4^ cell/well and allowed to attach overnight. CoCl_2_ (400 μM) was co-administered along with either non-encapsulated and liposome encapsulated STS for 24 h. Afterwards, media was collected and stored at −40 °C until the assay was performed.

### 2.11. Cell Migration Assay

The scratch assay was used to evaluate cell migration after a scratch/wound was performed. Briefly, EA.hy926 were plated in 24 well plates at 5.0 × 10^4^ cell/well and allowed to reach 100% confluency. After, a scratch/wound was performed using a sterile 200 μL tip (time 0 h). Cells were washed twice with warm PBS to remove debris. CoCl_2_ (400 μM) was co-administered along with either non-encapsulated or liposome encapsulated STS and images recorded using a Nikon Eclipse Ti-E phase-contrast inverted microscope (Nikon Instruments Inc., Melville, NY, USA) at 4 and 24 h. Wound areas were calculated using the image J wound healing tool and expressed relative to 0 h.

### 2.12. Capillary-Like Tube Formation Assay

Human umbilical vein endothelial cells (HUVEC) sourced from Promocell (Heidelberg, Germany) were routinely cultured in complete growth media (EGM-2) (Promocell, Heidelberg, Germany). For angiogenic assays, HUVEC were plated at a density of 1.0 × 10^4^ cells/well in 96-well plates previously coated with reduced growth factor Matrigel (Corning, NY, USA) and let attach for 1 h. Following, media was replaced with CoCl_2_ (400 μM) and either a non-formulated or liposome formulated STS dissolved in DMEM containing 0.5% FBS. The formation of capillary-like structures was assessed after 6 h. Images captured per well at 4× magnification, using a Nikon Eclipse Ti-E phase-contrast inverted microscope (Nikon Instruments Inc., Melville, NY, USA) and total branching length analysed using Image J Angiogenesis Analyzer tool from three independent experiments performed in duplicate.

### 2.13. Mitochondrial and Glycolytic Function

Oxygen consumption rates (OCR) and extracellular acidification rates (ECAR) were analysed in real-time using an XF24 Extracellular Flux Analyser (Agilent Technologies, California, USA). Briefly, EA.hy926 were plated is special V7 plates at density of 5.0 × 10^4^ cells/well (Agilent Technologies, California, USA) using standard DMEM growth media and allowed to attach for 24 h. Afterward, cells were treated with CoCl_2_ and either non-formulated or liposome-formulated STS and incubated at 37 °C and 5% CO_2_ atmosphere for 24 h. Following, culture media was changed to a non-buffered DMEM media (containing glucose 10 mM, pyruvate 1 mM and glutamine 2 mM) to allow temperature and pH equilibrium. Thereafter, OCR and ECAR were measured simultaneously three times to establish baseline measurements. Following, to evaluate mitochondrial and glycolytic function, a set of drugs/inhibitors: oligomycin (O) (1 mM) (Sigma-Aldrich, Dorset, UK), carbonyl cyanide 4-(trifluoromethoxy) phenylhydrazone (FCCP) (0.5 mM) (Sigma-Aldrich, Dorset, UK), 2 deoxy glucose (2-DG) (Sigma-Aldrich, Dorset, UK) and a mixture of rotenone and antimycin A (Rot/AntA) (1 mM) (Cayman Chemicals, Michigan, USA) were injected sequentially, using available ports to: inhibit the ATP synthase, uncouple oxidative phosphorylation, inhibit glycolysis and estimate non-mitochondrial respiration, respectively. From OCR readings, this experiment measures six parameters of the mitochondrial function: basal oxygen consumption, ATP-linked oxygen consumption, proton leak, maximal oxygen consumption, reserve capacity, and non-mitochondrial oxygen consumption as we have previously optimised [37,38,39] (Appendix A). From ECAR readings, the experiment allowed to calculate three glycolysis related parameters: basal glycolysis, glycolytic capacity and glycolytic dependence (Appendix A). After the completion of the determinations, OCR and ECAR measurements were normalized to protein content by the Bradford method.

### 2.14. ATP Levels

Levels of ATP were performed using cell culture supernatant after 24 h of exposure to treatments with CoCl_2_ and either non-formulated or liposome-formulated STS, using the ATP determination kit (A22066) from Molecular Probes, Oregon, USA and following the manufacturer’s instructions. 

### 2.15. Statistical Analysis

All results are presented as mean ± standard deviation (SD). Two replicates of at least three independent studies were used for all studies, unless stated otherwise. For multi-well plate cells assays, replicates of six were used for each experimental condition, repeated three times. When two groups were compared, an unpaired *t*-test was used. A two-way ANOVA was used to determine any statistically significant difference between means of three or more groups with a post-hoc Tukey’s multiple comparisons test applied to evaluate differences between groups. A *p* value ≤ 0.05 was considered statistically significant. All calculations were completed on Graphpad 8 (GraphPad Inc., La Jolla, CA, USA).

## 3. Results

### 3.1. Influence of Cholesterol and DOTAP on Liposome Characteristics

Our first approach aimed to assess the influence of lipid ratios on characteristics of liposome formulations whilst exploring these effects on STS entrapment. Lipid content of the liposome formulation was fixed at 10 mg/mL, to accommodate variations in cholesterol and DOTAP, the ratio of PC was modified as appropriate (Table 1). The ratio of cholesterol was increased from 7.5 to 15% of the lipid content and the ratio of DOTAP was increased from 7.5 to 22.5% of the formulation. DOTAP is a positively charged lipid and was selected to increase entrapment of the negatively charged thiosulphate ion. Neither the increase in cholesterol or DOTAP resulted in a significant change in either liposome size or polydispersity (Figure 1A,B). However, an increase in cholesterol ratio caused a significant reduction in zeta potential, whereas the increase in DOTAP ratio observed a significant increase in the zeta potential (Figure 1C). When the cholesterol was fixed at 7.5%, the zeta potential increased from 6.21 ± 1.68 to 24.13 ± 4.39 mV for liposomes formulated with a ratio of 7.5 to 22.5% DOTAP. When the cholesterol was fixed at 15%, the zeta potential increased from 3.02 ± 1.44 to 18.83 ± 3.29 mV for liposomes formulated with a ratio of 7.5 to 22.5% DOTAP. The entrapment efficiency of STS decreased significantly between liposomes formulated with 7.5% cholesterol compared to those formulated at 15% (*p* < 0.0001) (Figure 1D). Particularly, when cholesterol was fixed at 7.5%, we observed a significant increase in STS entrapment between formulations fixed with 22.5% compared to those fixed at 7.5% DOTAP from 22.55 ± 1.74 to 27.04 ± 1.05% (*p* < 0.05) (Figure 1D). However, when cholesterol was fixed at 15%, we did not observe any significant changes in STS entrapment whilst DOTAP increased. When comparing entrapment of STS between the formulation containing 7.5 and 15% cholesterol, we evidenced that when DOTAP is fixed at 7.5%, the entrapment of STS is reduced from 22.55 ± 1.74 to 17.64 ± 2.09% (*p* < 0.0001). Consistently, when DOTAP was fixed at 12.5 and 22.5%, respectively we evidenced a similar tendency to decrease STS entrapment (Figure 1D).

### 3.2. Cellular Uptake of STS

We next tested whether liposomes encapsulating STS were able to modulate the uptake of STS in cells (Figure 2A). The cellular uptake of STS encapsulated and non-encapsulated in liposomes formulated with 7.5% cholesterol and 7.5–22.5% DOTAP was measured over the course of 24 h using HPLC-UV analysis (Figure 2A). All dilutions corrected for any loss in the liposomal washing stage following HPLC-UV detection of STS to ensure the same concentration of STS was added to all cells. Uptake of liposomal cargo was visualised by fluorescence microscopy using formulating parameters described in Table 1—formulation 6 (Figure 2B). After 2 h, we observed that non-encapsulated STS allowed 87.96 ± 3.14% of STS to diffuse into equilibrium within EA.hy926 cells. We also evidenced that liposome encapsulated STS provided a maximum diffusion of STS of 56.60 ± 1.46% when DOTAP was fixed at 22.5%. Overall, as the DOTAP loading increased from 7.5 to 22.5%, the content of STS detected within the cells increased, with DOTAP fixed at 22.5% showed a significant increase in STS uptake when compared to formulations with fixed DOTAP at 7.5% (*p* < 0.01). After 4 h, there was a slight decrease in non-encapsulated thiosulphate that had diffused into equilibrium within the cells (80.71 ± 4.55%). As the DOTAP loading increased from 7.5 to 22.5%, the content of STS detected within the cells increased from 52.56 ± 2.78 to 56.56 ± 2.47%; however, these differences were not statistically significant. After 24 h, non-encapsulated STS diffused into equilibrium within the cells was at 57.57 ± 3.08% of the maximum expected concentration. At this time point, we evidenced a slight significant difference between the STS content in cells exposed to non-encapsulated or encapsulated STS at any fixed DOTAP ratio (*p* < 0.001). Interestingly, when compared the uptake of STS at 24 with those levels measured at 2 and 4 h, we evidenced that non-encapsulated STS displayed significantly lower levels than those measured at 2 h (*p* < 0.001). However, we observed that after 24 h, the uptake of STS from formulations was higher than those applied for 2 or 4 h. This observation suggest that non-encapsulated STS may be metabolised at a higher rate when compared to their formulation counterparts and suggest that liposome formulations not only offer a have a slower rate of uptake over 2–4 h but may potentially prevent rapid metabolism of STS in media which may translate to sustained biological action.

### 3.3. H_2_S Release from Liposomal Formulated and Non-Formulated STS

Following our results demonstrating that DOTAP ratio was not influencing STS uptake at 24 h, we prepared liposome formulations at fixed DOTAP 7.5%. Encapsulated and non-encapsulated STS were prepared at a final concentration of 0.75 mg/mL of STS. We next assessed the release of H_2_S into the media from EA.hy926 cells over the course of 24 h (Figure 3). Whilst the peak of H_2_S release for the non-encapsulated formulation was at 2 h, a delayed peak was observed for encapsulated STS at 4 h. At peak release times, the concentration of H_2_S released from non-encapsulated STS was 11.95 ± 0.73 µM, significantly higher than that of the encapsulated STS at 8.86 ± 0.62 µM (*p* < 0.001), suggesting controlled release properties of the formulated STS.

### 3.4. Formulation of STS into Liposomes Retain Pro-Angiogenic Properties of STS

Once we determined that our liposome formulation was successfully up-taken by cells and releasing controlled levels of H_2_S, we aimed to evidence whether H_2_S properties were retained. In scenarios of vascular hypoxia, H_2_S has been demonstrated to exert cytoprotective and pro-angiogenic effects [40,41]. Therefore, to explore whether STS encapsulated in liposomes was able to retain these protective properties, we first established a hypoxia-like environment by exposing EA.hy926 to CoCl_2_ (400 μM), resulting in upregulation of HIF-1α and VEGF (isoform 165) mRNA levels and elevated VEGF protein levels (Appendix A). Once hypoxic signals were confirmed in our CoCl_2_ model, we co-exposed EA.hy926 to CoCl_2_ and either liposome encapsulated or non-encapsulated STS and investigated pro-angiogenic signals (Figure 4). 

The administration of liposomal STS resulted in further VEGF^165^ and Glut-1 mRNA increase when compared to non-formulated STS in hypoxia (*p* < 0.05 and *p* < 0.001, respectively) (Figure 4A,C). Consistently, the determination of VEGF levels in tissue culture media indicated that co-exposure of CoCl_2_ with formulated STS significantly stimulated the release of VEGF (*p* < 0.01) in comparison to those exposed to non-formulated STS and CoCl_2_ (Figure 4B).

Next, we evidenced that liposomal STS enhanced pro-angiogenic signals in hypoxia-like environment, thus, we explored the potential of liposome encapsulated STS to retain pro-angiogenic properties by exploring wound closure at 4 and 24 h after the scratch/wound was performed. Figure 4E shows that at 4 h, there is no difference in the percentage of wound closure between non-formulated and formulated STS. However, at 24 h we evidenced that liposomal STS enhanced wound closure in comparison to their non-formulated counterpart (*p* < 0.01). 

Furthermore, we exposed HUVEC to CoCl_2_ and either non-encapsulated and liposomal STS and measured capillary-like formation. We evidenced that formulated STS further enhanced tube formation measured as total branching length, in comparison to cells exposed to non-encapsulated STS in hypoxia (*p* < 0.0001) (Figure 4D,F). These observations suggest that liposomal STS was able to retain the pro-angiogenic effect of STS.

### 3.5. Liposomal STS Promotes Mitochondrial Function under Hypoxia

Recently, the use of H_2_S donors has been recognised as a potential approach to modulate cellular bioenergetics and ATP production in a myriad of adverse pathological conditions [37,42,43,44,45,46]. Furthermore, as others have previously reported, under hypoxia, cells fail to activate mitochondrial respiration, relying primarily on glycolysis for energy supply [47,48]. However, recent evidence has suggested the potential of H_2_S donors to modulate mitochondrial dysfunction in hypoxia [10,49]. Thus, we explored whether liposomal STS may retain mitochondrial protective properties in our hypoxia model. We assessed mitochondrial respiration parameters in real-time using a XF24 Seahorse Analyser by measuring the oxygen consumption rates (OCR) by time after the sequential administration of drugs/inhibitors oligomycin, FCCP, 2-DG and mixture of rotenone and antimycin A in EA.hy926 co-exposed to CoCl_2_ and either non-encapsulated or formulated STS for 24 h (Figure 5A).

These OCR determinations allowed the calculation of parameters of mitochondrial function (Appendix A) as evidenced in Figure 5B. As expected, our model of hypoxia with CoCl_2_ significantly reduced the basal respiration (*p* < 0.0001), maximal respiration (*p* < 0.0001), spare respiratory capacity (*p* < 0.0001) and ATP-linked OCR (*p* < 0.0001) when compared to non-treated cells. Co-exposure of CoCl_2_ with either non-encapsulated STS or liposomal STS formulation demonstrated an improvement of parameters of the mitochondrial function in comparison to cells treated with CoCl_2_ only (basal respiration (*p* < 0.05), maximal respiration (*p* < 0.05), spare respiratory capacity (*p* < 0.05) and ATP-linked OCR (*p* < 0.05)). Interestingly, treatment measured in the presence of liposomal STS significantly improved the maximal respiration (*p* < 0.0001) and spare respiratory capacity (*p* < 0.0001) when compared to those cells co-exposed to CoCl_2_ and non-encapsulated STS (Figure 5B). 

We also explored the effect of CoCl_2_ and STS on the production of ATP. CoCl_2_ significantly enhanced the production of ATP in comparison to non-treated cells (*p* < 0.0001). Interestingly, we observed that both non-encapsulated and liposomal STS reduced ATP levels when compared to cells exposed to CoCl_2_ only. However, only liposomal STS induced a significant reduction in ATP levels when compared to CoCl_2_ only treated cells (*p* < 0.05) (Figure 5C). H_2_S has previously been recognised as a modulator of redox responses by increasing the expression of antioxidant genes such as thioredoxin [50]. We explored whether liposome encapsulating STS may retain these effects by evaluating the mRNA expression of thioredoxin in cells exposed to CoCl_2_. Figure 5D shows that liposome encapsulating STS significantly increased thioredoxin mRNA when compared to cells exposed to CoCl_2_ only.

### 3.6. Liposomal STS Promotes Cellular Metabolic Switch Favouring Mitochondrial Function in Hypoxia-Like Environment

The hypoxia-like environment induced by CoCl_2_ led to a cellular metabolic switch enhancing the dependence on glycolytic pathways (Figure 6). As Figure 6A shows, the administration of CoCl_2_ resulted in higher extracellular acidification rates (ECAR) measured by time. However, co-exposure with either non-encapsulated STS or liposomal STS formulation led to lower ECAR measurements (Figure 6A). Using data obtained from Figure 6A, we calculated parameters of the glycolytic pathway; basal glycolysis, glycolytic capacity and glycolytic reserve (Figure 6B).

Here we evidenced that our hypoxia-like model enhanced the use of glycolysis as evidenced by approximately a two-fold increase in basal glycolysis in comparison to non-treated cells. In addition, we evidenced that both non-formulated and liposomal STS abrogated basal glycolysis in comparison to those cells treated with CoCl_2_ only (*p* < 0.05). When we analysed the glycolytic capacity, which represents the difference resulting from maximum ECAR rates after oligomycin injection and basal glycolysis (Appendix A) we observed that treatment with CoCl_2_ significantly enhanced the ability of cells to use glycolysis as a source of energy (*p* < 0.01) vs. control and these effects were sustained in those cells co-exposed with non-encapsulated STS (*p* < 0.001) vs. control. Interestingly, those cells co-exposed to CoCl_2_ and formulated STS showed similar glycolytic capacity as non-treated cells (Figure 6B). Moreover, when we examined the glycolytic reserve, which resembles the ability of cells to rely on glycolysis (measured as the difference between maximum ECAR after 2-DG and basal glycolysis), we evidenced that similar to the glycolytic capacity, only those cells exposed to CoCl_2_ and liposomal STS showed similar glycolytic reserve as non-treated cells (Figure 6B).

Together Figure 5 and Figure 6, exploring both the mitochondrial function and glycolytic pathways, suggest that formulation of STS encapsulated into liposomes retain H_2_S abilities to enhance the mitochondrial function and therefore to favour the reliance on mitochondria for energetic purposes when oxygen supply is limited.

## 4. Discussion

This study explored the feasibility of formulating the H_2_S donor, STS, into liposomes and observed their potential for controlling the release of H_2_S whilst sustaining biological activity of the compound in hypoxia-like settings. Here, we designed, developed and characterised cationic liposome formulations and quantified STS entrapment, H_2_S release and confirmed that this delivery system retains biological properties of H_2_S by enhancing pro-angiogenic signals and promoting mitochondrial function impaired in hypoxia. These novel findings position liposome-based nanoparticles as suitable carriers for H_2_S donors and highlights their potential translation to the clinic due to their safety, improved rates of delivery and sustained biological effects.

Over the past two decades, research into H_2_S as a therapeutic option has gained particular interest due to its involvement in the development of human pathophysiological conditions including: neurodegeneration, inflammation, metabolism and cardiovascular-related disorders [51]. Jiang et al. reported low levels of H_2_S in plasma from patients suffering from coronary heart disease [52] whilst others have reported H_2_S levels are also reduced in patients displaying heart failure [17]. Not surprisingly, animal models of cardiovascular disease have reported a defect in the endogenous generation of H_2_S in these settings [53]. These observations have supported investigation of the potential of H_2_S donors to modulate these adverse conditions [54,55]. In human physiology, one of the intermediate endogenous metabolites of H_2_S in the nonenzymatic pathway is thiosulphate [56,57]. Thiosulphate is an intermediate in the sulphur metabolism pathway from cysteine where formation of the H_2_S results from a reductive reaction involving pyruvate, a hydrogen donor [58]. In addition to thiosulphate being generated by H_2_S in sulphide oxidation pathways [58,59], under hypoxic conditions, thiosulphate itself generates H_2_S [60]. In this regard, STS, has been investigated in patients with acute coronary syndrome undergoing coronary angiography, showing that the administration of two doses (15 g each), 6 h apart of STS had a therapeutic benefit in this patient cohort [26]. However, delivery challenges associated with H_2_S donors, such as fast H_2_S release rates, extensive dosage regimes, and potential toxicity when present in excess have limited clinical use. Therefore, we proposed a novel liposomal STS delivery system may provide many advantages such as reducing active compound breakdown efficacy, prolonged circulation times, controlling and prolonging H_2_S release to mimic basal production, along with having potential for targeted delivery [30,61]. 

Defining liposome characteristics such as size, polydispersity, charge and drug entrapment efficiency is necessary to ensure reproducible drug delivery rates and therefore suitability as drug delivery systems. A liposome preparation homogenous in size is critical as this will influence tissue distribution in vivo in addition to affecting drug release kinetics. In this study, a desirable polydispersity (up to 0.3) was determined, indicative of a liposomal formulation homogenous in size [62]. As previously described, depending on the payload of interest and destination target, liposomal complexes can be designed to have a neutral, positive or negative overall charge [63]. In this regard, cationic liposomes have been established as effective carriers of negatively charged nucleic acids and are currently being used as gene delivery systems [61,64].

Consistent with previous reports [65], we observed that an increase in the DOTAP content within the liposomal bilayer caused an increase in charge as defined by zeta potential. This was also reflected in a decrease of cholesterol content. DOTAP is a cationic lipid and thus imparted a positive charge to the liposomes, allowing for the greater association with STS. Consistent with our observations displaying higher STS entrapment rates dependent on DOTAP content, a study exploring the effects of DOTAP content in the encapsulation efficiency of a negatively charged H_2_S donor (DTO) showed that liposomes formulated with 3 mol % of DOTAP led to a significantly higher encapsulation efficiency compared to liposome formulated without DOTAP [66]. Interestingly, we did not observe this trend in the presence of higher cholesterol rates (15%). Cholesterol is a neutral lipid, and thus we infer these observations might be related to a slight decrease in the liposomal charge leading to reduced STS adherence when the cholesterol ratio increased. Furthermore, membrane hydrophobicity is determined by cholesterol content [67] thus the thiosulphate ion would be more likely to be bound to a membrane of lower hydrophobicity. While studies have shown the ability of cholesterol to effectively package within a liposomal membrane can improve bilayer stability, it may also counteract the drug encapsulation efficiency within the liposome [68,69,70]. Our data suggest that the interaction of STS with the liposome was ionic and thus adsorbed onto the bilayer of the liposome.

During the course of 24 h, all liposomal formulations were able to slow the cellular uptake of STS. Within the first 2 h, liposomes formulated with 7.5% DOTAP had the lowest cellular delivery of STS. Cellular particulate uptake is dependent upon influences such as particle charge and affinity [71,72,73]. Likewise, particle surface charge influences the interaction with cell membrane. In this regard, the cell membrane surface is dominated by negatively charged sulphated proteoglycans molecules critical in cellular proliferation and migration [74,75]. These molecules are associated with anionic glycosaminoglycan side chains (heparan, dermatan, keratan or chondrotine sulfates) and interaction between proteoglycans and cationic liposomes tend to be ionic [76]. Liposome in vitro cellular uptake has been established to be influenced by surface charge with anionic and cationic liposomes being better internalized than those that are neutral [77]. Similar to previous studies in which increasing percentages of cationic lipid, DOTAP, enhanced cellular uptake [78,79], our study observed that liposomes with a greater DOTAP ratio, and thus a greater surface charge, displayed a faster cellular uptake, suggesting an ionic interaction. A study comparing the uptake of cationic and anionic liposomes observed the former had improved cellular uptake, postulating that the anionic formulation could have had an electrostatic repulsive-force with the cellular membrane [80]. This is in tandem with a study applying cationic liposomes in oligonucleotide delivery to immortalized human keratinocyte cells observing liposome uptake within 24 h [81]. Furthermore, research developing chemotherapy against malignant melanoma using mouse B16 melanoma cells as well as Normal Human Dermal Fibroblasts observed a greater uptake of cationic liposomes by cells in the injection site compared with neutral liposomes due to the electrostatic interaction with the negative-charged cell membrane [82]. This similarity observed with our results suggest that electrostatic interactions between STS-loaded liposomes and the negatively charged coat of membrane lipids may be responsible for the enhanced uptake of STS when formulated with higher ratios of DOTAP.

One of the main drawbacks in the translation of H_2_S-based compounds to the clinic is their rapid degradation. In the case of STS, oral administration is not appropriate due to its quick degradation in the stomach [83]. Furthermore, many reports in vivo administering H_2_S-based compounds intraperitoneally require intense regimes of administration, with daily injections required for periods of days to weeks to observe biological effects [84,85]. Critically, we observed the controlled release of H_2_S from liposomal encapsulated STS compared with non-encapsulated STS suggesting the liposomes were able to entrap STS and delay its release thus delaying the generation of H_2_S. This is crucial in the delivery of H_2_S donors and suggest the possibility of better translation to the clinic, offering patient-friendly regimes. In vivo experimentation is needed to establish the ability of the liposomes to delay H_2_S release in vivo, liposomes may require further optimisation for a clinically relevant sustained H_2_S release profile. 

The exploration of STS as a potential therapeutic approach has been explored extensively and it has been proven that exogenous thiosulphate administered during hypoxia, induces H_2_S metabolite bioavailability in HUVEC [86]. Thiosulphate can modulate the availability of H_2_S metabolites and signalling under various culture conditions that impact angiogenic activity [86]. In order to confirm that our delivery system retained STS beneficial effects, we assessed their effects in sustaining pro-angiogenic responses in chemically induced hypoxia exerted by CoCl_2_. Liposome-based delivery improved wound closure whilst enhancing capillary-like formation in a hypoxia-like environment. As reported by Liu et al. (2010) the H_2_S donor NaHS promotes endothelial cells’ proliferation and migration in hypoxia [16]. These observations align with our study as we consistently report an increase in mRNA and protein expression of pro-angiogenic VEGF isoform, 165, consistent with Liu et al. (2010). Additionally, in a model of unilateral, permanent femoral artery ligation model of hind-limb ischemia established in C57BL/6J wild-type and endothelial nitric oxide synthase–knockout mice, saw the administration of sodium sulphide restored chronic ischemic tissue function while increasing markers of angiogenesis. These studies further support the potential of STS in the modulation of angiogenesis in oxygen deprived tissues whilst our novel controlled-released approach may provide opportunities for reducing dosage regimes.

In an attempt to address the mechanism by which STS-loaded liposomes might enhance angiogenesis, we explored the potential for this novel formulation to modulate cellular bioenergetics and ATP production in a hypoxia-like environment. As previously established, the limited availability of oxygen during hypoxia forces cells to rely on less efficient metabolic routes such as glycolysis for ATP production [47,48]. Nonetheless, previous reports in our laboratory and others suggest that H_2_S donors enhance mitochondrial bioenergetics [37,46,87] and modulate anti-oxidant [87] and anti-inflammatory processes in the vascular system [26,40,88,89], potentially by acting as an electron donor to the electron transport chain [90]. Figure 5 and Figure 6 highlight that non-encapsulated and liposome formulated STS significantly enhanced rates of mitochondrial respiration in cells challenged with hypoxia. Taken together with the cellular uptake data suggesting non-encapsulated STS was also metabolised fastest with lower amounts present in the cell, these data infer liposome-loaded STS provided sustained effects when compared to those exposed to non-encapsulated STS. These data suggest that liposomal formulations of STS may play a role in reducing the dosing regimen in clinical settings, such as in patients with acute coronary syndrome undergoing coronary angiography [26]. Moreover, we were able to show that our novel liposome formulation encapsulating STS had a significant effect in the production of ATP and antioxidant protection by upregulating thioredoxin mRNA levels. Our results are consistent with previous reports showing the potential for H_2_S donors to modulate mitochondrial dysfunction during hypoxia [10,49] and enhance antioxidant genes expression [50] demonstrating that liposome encapsulation of STS retains its ability to modulate cellular bioenergetics redox defences whilst sustaining these effects when compared to the non-formulated H_2_S donor.

To the best of our knowledge, this study is the first to suggest the potential of controlled release liposome-based formulations encapsulating H_2_S donors, such as STS, which could potentially reduce therapeutic STS dosing. Further in vivo experiments are needed to establish whether this approach can be translated into clinic to offer a more cost-effective and patient-friendly alternative with a reduced dose frequency compared to currently available treatment options for ischemic cardiovascular diseases.

## 5. Conclusions

Cardiovascular diseases such as ischemic heart disease and cardiomyopathy represent a major global health problem and are a leading cause of morbidity and mortality worldwide. This highlights the ongoing need for an effective therapeutic approach to inhibit defective ischemic molecular mechanisms and promote new blood vessel formation to restore diminished blood supply. Hydrogen sulphide generating donors such as STS have been shown to improve cardiac function after myocardial ischemia and suggested as a treatment for acute coronary syndrome. However, H_2_S doners are limited in clinical application due to poor biodistribution, high tissue clearance and the rapid release of H_2_S. To address this therapeutic challenge, we designed and optimised a liposomal formulation to encapsulate STS and explore its potential therapeutic effect in vitro. Liposomes containing cationic lipid DOTAP were successfully loaded with STS and exhibited, controlled cellular uptake and the release of STS, as well as H_2_S. In the hypoxic state, liposomal STS not only induced a pro-angiogenic response in human endothelial cells hallmarked by the enhanced capillary tube formation and wound closure but also improved the mitochondrial bioenergetics profile. These promising results suggest liposomal STS as a potential strategy for the treatment of ischemic cardiovascular diseases. Future studies are required to confirm therapeutic liposomal STS effects in vivo and to explore a potential transition into clinical use.

## Figures and Tables

**Figure 1 antioxidants-11-02092-f001:**
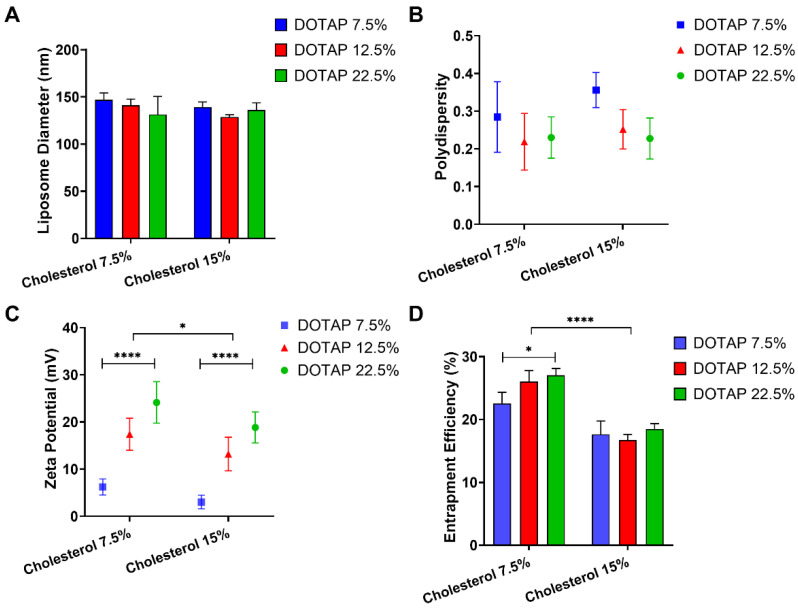
The effects of DOTAP and cholesterol ratios within liposomal composition on size, polydispersity, charge and STS entrapment. Liposomes formulated with DSPE-PEG and varying ratios of PC, DOTAP and cholesterol were prepared by the extrusion method. Sodium thiosulphate (STS) was dispersed within the buffer (25 mg/mL). (**A**) Liposome size distribution and (**B**) polydispersity was determined by DLS, (**C**) zeta potential was determined by photon correlation spectroscopy using a Zetaplus and (**D**) entrapment efficiency determined by HPLC-UV. Data represents mean ± SD. (*n* = 3) independent batches. Statistical differences: * *p* < 0.05 and **** *p* < 0.0001.

**Figure 2 antioxidants-11-02092-f002:**
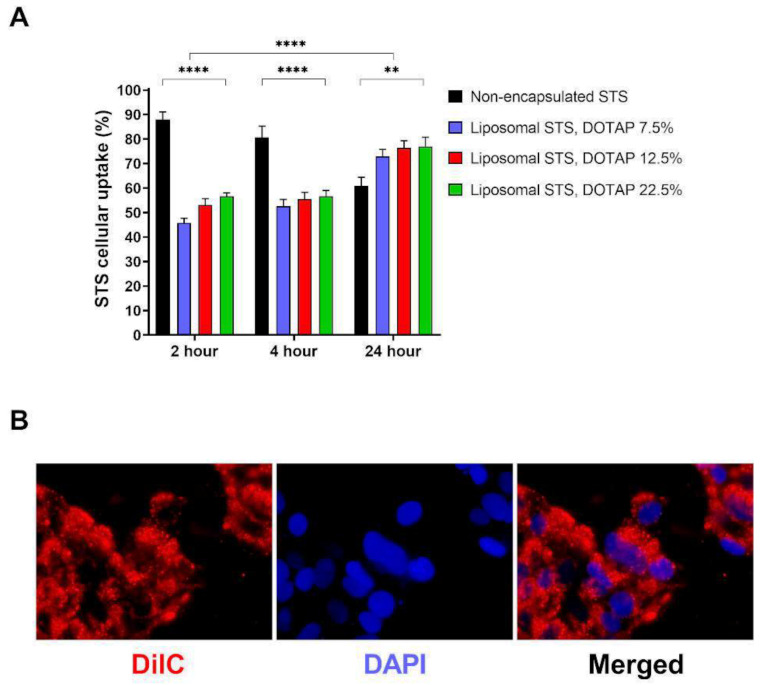
The rates of STS cellular uptake over time. Formulation was applied to EA.hy926 cells and cell lysate was used to measure uptake after 2, 4 or 24 h exposure using HPLC-UV analysis. (**A**) Cellular uptake of STS in media compared with liposomal formulations consisting of varying ratios of PC, cholesterol, DSPC-PEG and DOTAP. (**B**) Cells were grown on coverslips and exposed to liposomes entrapping the fluorescent probe DilC (red). Cell nuclei was visualised using DAPI (blue). Data represents mean ± SD (*n* = 6) independent batches. Statistical differences: ** *p* < 0.01 and **** *p* < 0.0001.

**Figure 3 antioxidants-11-02092-f003:**
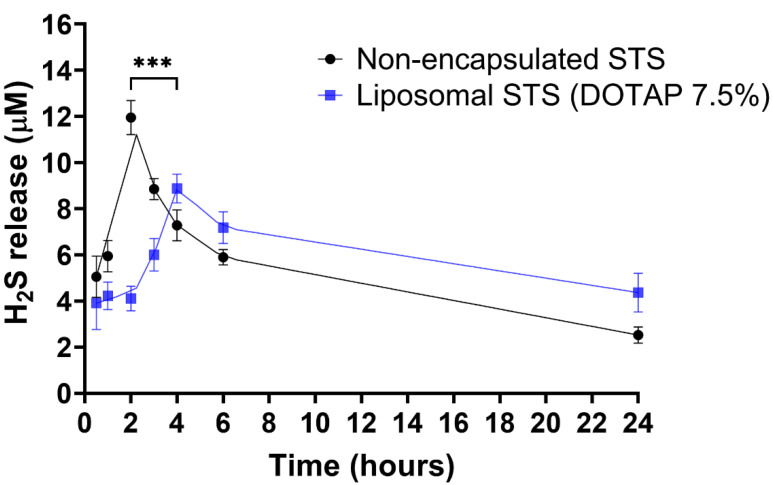
H_2_S release from non-formulated and liposome encapsulating STS. Hourly H_2_S release values are plotted with curve-fitting results to highlight the donor compound decomposition. Statistical differences: *** *p* < 0.001 between the maximal points of H_2_S released from the non-formulated and liposome formulation.

**Figure 4 antioxidants-11-02092-f004:**
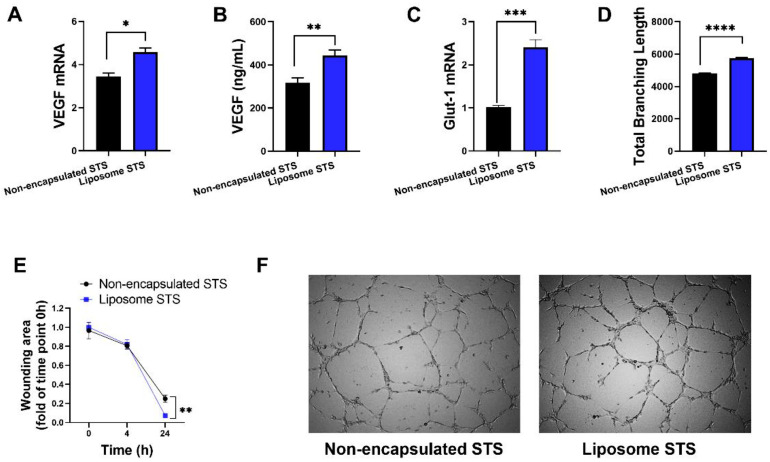
Liposomal STS maintains pro-angiogenic signals of H_2_S under conditions mimicking hypoxia established by CoCl_2_. EA.hy926 were exposed to CoCl_2_ (400 μM) in combination with either non-formulated or liposome formulated STS for 24 h. (**A**) Expression of VEGF mRNA measured by qPCR. (**B**) VEGF release measured by ELISA in cell culture media. (**C**) Expression of Glut-1 mRNA measured by qPCR. (**D**) Capillary-like formation measured as total branching length was investigated in HUVEC exposed to CoCl_2_ (400 μM) in combination with either non-formulated or liposome formulated STS and calculated using Image J angiogenesis tool. (**E**) Cell migration measured as percentage of wound closure relative to 0 h. (**F**) Images representative of tube formation assays captured at 4×. Data represents mean ± SD (*n* = 3). Statistical differences: * *p* < 0.05, ** *p* < 0.01, *** *p* < 0.001, **** *p* < 0.0001.

**Figure 5 antioxidants-11-02092-f005:**
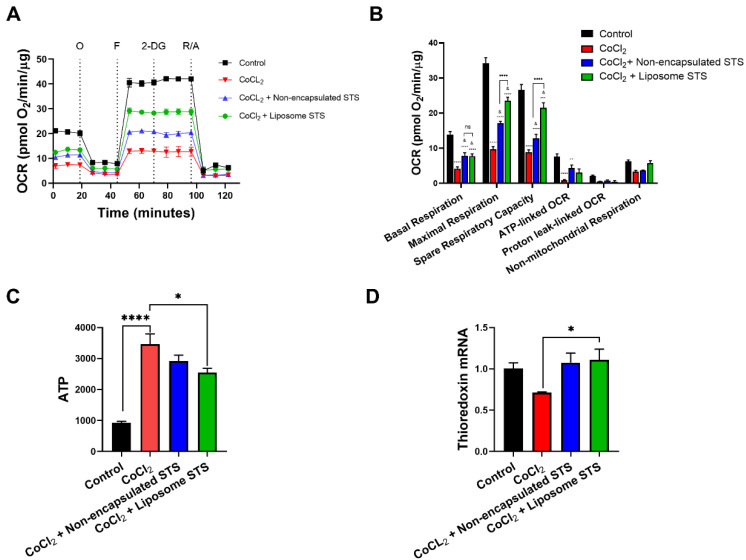
Liposomal STS enhanced mitochondrial bioenergetics under conditions mimicking hypoxia. EA.hy926 were exposed to CoCl_2_ (400 μM) in combination with either non-formulated or liposomal STS for 24 h. Afterward, mitochondrial function was evaluated in real-time using a XF24 Seahorse analyser and ATP levels measured by luminescence. (**A**) Traces of oxygen consumption rates (OCR) expressed by time after the sequential administration of drugs/inhibitors: oligomycin (O), FCCP (F), 2- deoxy glucose (2-DG) and mixture of rotenone and antimycin A (R/A). (**B**) Calculated mitochondrial function parameters from graph A. (**C**) ATP levels. (**D**) Expression of thioredoxin measured by qPCR. (*n* = 5). Statistical differences: (**B**) ** *p* < 0.01, *** *p* < 0.01, **** *p* < 0.001 vs. control, & *p* < 0.05 vs. CoCl_2_. (**C**,**D**): * *p* < 0.05, **** *p* < 0.001.

**Figure 6 antioxidants-11-02092-f006:**
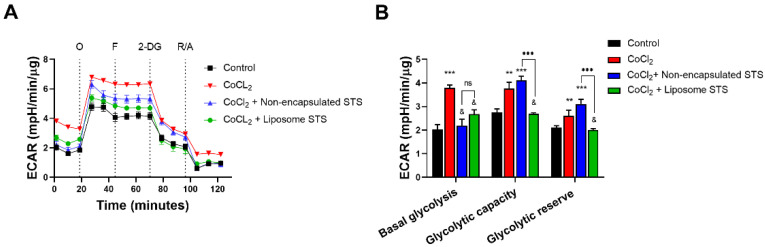
Liposomal STS abrogated glycolytic dependence of endothelial cells under conditions mimicking hypoxia. EA.hy926 were exposed to CoCl_2_ (400 μM) in combination with either non-formulated or liposome formulated STS for 24 h. Afterward, extracellular acidification rates (ECAR) was evaluated in real-time using a XF24 Seahorse analyser. (**A**) ECAR traces expressed by time after the sequential administration of drugs/inhibitors: oligomycin (O), FCCP (F), 2- deoxy glucose (2-DG) and mixture of rotenone and antimycin A (R/A). (**B**) Calculated glycolytic function parameters from graph A. (*n* = 5). Statistical differences: ** *p* < 0.01, *** *p* < 0.01 vs. control, & *p* < 0.05 vs. CoCl_2_.

**Table 1 antioxidants-11-02092-t001:** The ratios of cholesterol and anchor lipids PC and DOTAP.

Formulation Number	PC	DOTAP	DSPC-PEG 2000	Cholesterol
1	57.5	22.5	5	15
2	47.5	12.5	5	15
3	52.5	7.5	5	15
4	65	22.5	5	7.5
5	75	12.5	5	7.5
6	80	7.5	5	7.5

## Data Availability

Data is contained within the article and Appendix A.

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
