# Peer review of "Sodium Thiosulphate-Loaded Liposomes Control Hydrogen Sulphide Release and Retain Its Biological Properties in Hypoxia-like Environment"

_antioxidants, 2022, doi:10.3390/antiox11112092_

Round 1

Reviewer 1 Report

  1. The study titled “Sodium thiosulphate-loaded liposomes control hydrogen sulphide release and retain its biological properties in hypoxia- 3 like environment” is promising. However, the reviewer suggests the following points to address for further enrichment of the study. 
  2. HUVECs are a good starting point for angiogenesis study. However, cell does not mimic the actual physiological scenario. HUVECs drastically change their biological properties if those are not under blood flow. Finally, the efficacy of any drug or delivery system should be checked in animal system. The reviewer, therefore, recommends the use of animal studies at least for cellular uptake of STS and HIF-1 mediated gene expressions. The authors may use hypoxia chamber for the induction of hypoxia in in vivo settings.
  3. H2S release is not very different between non-encapsulated and liposomal STS except the peak release. The authors should explain this. 
  4. The reviewer thinks that all the figures should be color coded instead of using different shades for different group.
  5. Other HIF-1 regulated genes should also be checked along with VEGF.
  6. Figure 4E is very dark and unreadable. The authors should replace it with high quality image.
  7. The reviewer could not find supplementary files which are mentioned in the manuscript.
  8. STS modulates cellular redox system which is important in cardiovascular pathophysiology (primary goal of the study). The authors should check the redox profile of the STS treated cells & animals. 
  9. There are some grammatical and spelling mistakes in the manuscript. It is advised that the authors should rewrite the manuscript with the help of a native English speaker.

Author Response

Response to Reviewer 1 Comments

We respectfully submit revisions, following peer-review for our article manuscript entitled: “Sodium thiosulphate-loaded liposomes control hydrogen sulphide release and retain its biological properties in hypoxia-like environment”.

Below, we addressed comments detailing the revisions in the manuscript and our responses to the reviewers' comments are highlighted in blue font.

1. The study titled “Sodium thiosulphate-loaded liposomes control hydrogen sulphide release and retain its biological properties in hypoxia- 3 like environment” is promising. However, the reviewer suggests the following points to address for further enrichment of the study. 

We would like to thank the reviewer for the time taken to review our manuscript. We really appreciate the reviewer commenting this work is promising. Moreover, the suggestions to improve our manuscript are insightful and we would like to thank for these observations. Bellow, we are addressing these comments.

2. HUVECs are a good starting point for angiogenesis study. However, cell does not mimic the actual physiological scenario. HUVECs drastically change their biological properties if those are not under blood flow. Finally, the efficacy of any drug or delivery system should be checked in animal system. The reviewer, therefore, recommends the use of animal studies at least for cellular uptake of STS and HIF-1 mediated gene expressions. The authors may use hypoxia chamber for the induction of hypoxia in in vivo settings.

We appreciate this observation. We indeed agree with the reviewer that having animal experiments are a fundamental step towards the broader application of our approach into the clinical scenario. We unfortunately are not in the capacity at this stage to explore these in vivo. As the reviewer has indicated, HUVEC and overall in vitro experiments do not resemble the response we may expect in vivo. In order to address this, we have made sure we inform of this limitation in our work by adding the following statement “Further in vivo experiments are needed to establish whether this approach can be translated into clinic to offer a more cost-effective and patient-friendly alternative with a reduced dose frequency compared with currently available treatment options for ischemic cardiovascular diseases” in section 4, page 16. We do hope this manuscript opens up opportunities to further collaborate with other research teams that are in the capacity of explore more in depth our observations in vivo and we anticipate that future work will also explore other aspects of the application of our formulations, such as pharmacodynamics.

3. H2S release is not very different between non-encapsulated and liposomal STS except the peak release. The authors should explain this. 

We appreciate this observation. We agree with the reviewer that the H2S release is not very different between non-encapsulated and liposomal STS, however, we did observe the peak release took twice as long. We believe liposomes may be optimized to alter this release profile, however, that was not within the scope of this work. We would like to investigate this in future in vivo experimentation. Therefore, in order to address this, we have made sure we inform of this limitation in our work by adding the following statement “In vivo experimentation is needed to establish the ability of the liposomes to delay H2S release in vivo, liposomes may require further optimisation for a clinically relevant sustained H2S release profile” in section 4, page 15.

4. The reviewer thinks that all the figures should be color coded instead of using different shades for different group.

We thank the reviewer for this observation to improve our manuscript and overall, the insights of our results. We have now modified the figures and added color coding for each experimental group.

5. Other HIF-1 regulated genes should also be checked along with VEGF.

We appreciate this suggestion. We are now adding new data in where we show the expression of Glut-1 as HIF-1α downstream regulated gene in Figure 4. Glut-1 expression is particularly relevant in our study as it is implicated in glucose metabolism, directly involved in hypoxia-like settings. Thus, labeling of figure 4 has changed. This can be evidenced in the manuscript in where figure 4C is now labeled 4D, 4D as 4E and 4E is now referred as 4F. These changes and reference to the addition of Glut-1 results have been added to section 2.9 (Quantitative RT-PCR) in the methods section, page 5. We have also added these important downregulated genes expressions to supplementary figure 2, supporting our hypoxia-like in vitro model.

6. Figure 4E is very dark and unreadable. The authors should replace it with high quality image.

We apologise for these images not being the best quality. We have now replaced these images with clearer and higher quality ones.

7. The reviewer could not find supplementary files which are mentioned in the manuscript.

We apologise these supplementary figures were not easily found by the reviewer. We noticed that the formatting of the manuscript draft is showing the legend of supplementary figure 1 in the next page (page 7 out of 20) and therefore is not clear. We hope the new formatting makes easier to read and find the supplementary information.

8. STS modulates cellular redox system which is important in cardiovascular pathophysiology (primary goal of the study). The authors should check the redox profile of the STS treated cells & animals. 

We thank the reviewer for this observation. Along with our results in Figure 5 and 6 in where we show the effects of liposome encapsulating STS in retaining STS effects on the mitochondrial and glycolytic function respectively, we are now able to show data regarding antioxidant genes mRNA expression in response to CoCl2 and STS. Our results in figure 5 and 6 show that CoCl2 disrupts the cellular bioenergetics whilst liposome encapsulating STS is able to abrogate these effects. Understanding the role of H2S as antioxidant molecule and the effects that H2S-donating compounds have in the transcription of Nrf2-dependent antioxidant gene thioredoxin (Calvert, 2009), we have added to figure 5 (Figure 5D) mRNA expression of thioredoxin. We observed that both non-encapsulated and liposome encapsulating STS restore mRNA expression of thioredoxin to similar levels observed in non-treated cells. Liposomal STS was statistically different to cells exposed to CoCl2. We infer these effects may be mediated by Nrf2, however, as this is not the goal of the study, we have not further this exploration. The reviewer may find this data added to the results and discussion section. Moreover, details of thioredoxin primer sequence were added to the methods section (section 2.9).

Calvert JW, Jha S, Gundewar S, Elrod JW, Ramachandran A, Pattillo CB, Kevil CG, Lefer DJ. Hydrogen sulfide mediates cardioprotection through Nrf2 signaling. Circ Res. 2009 Aug 14;105(4):365-74. doi: 10.1161/CIRCRESAHA.109.199919. Epub 2009 Jul 16. PMID: 19608979; PMCID: PMC2735849.

9. There are some grammatical and spelling mistakes in the manuscript. It is advised that the authors should rewrite the manuscript with the help of a native English speaker.

We thank the reviewer for this suggestion. We have edited the manuscript in order to comply with the grammatical misspellings highlighted by the reviewer taking advantage that several of our co-authors are native English speakers.

Reviewer 2 Report

The paper presents information regarding “Sodium thiosulphate-loaded liposomes control hydrogen sulphide release and retain its biological properties in hypoxia-like environment”.

The manuscript is clearly presented, well documented and the subject is of interest. In order to

improve the quality of the manuscript I recommend the authors:

1.      The authors should explain the novelty of the work with more details.

2. Figure 5 and Figure 6: The text on the axes must be enlarged. It is not clearly visible.

3. The grammatical and typo errors should be revised:

Change CoCl2 to CoCl2

Change RPM to rpm.

…temperature.  Coverslips is modified with … temperature. Coverslips …

Change H2S to H2S.

The conclusion should be a summarized version of the total manuscript.

 It is better to summarize what the author has discussed in the manuscript briefly. 

I recommend publication of this paper after minor revision.

Author Response

Response to Reviewer 2 Comments

We respectfully submit revisions, following peer-review for our article manuscript entitled: “Sodium thiosulphate-loaded liposomes control hydrogen sulphide release and retain its biological properties in hypoxia-like environment”.

Below, we addressed comments detailing the revisions in the manuscript and our responses to the reviewers' comments are highlighted in blue font.

We thank the reviewer for the time taken to revise our manuscript and to suggest these very valuable insights to improve our manuscript.

  1. The authors should explain the novelty of the work with more details.

We are thankful for this observation. In order to highlight the novelty of our results more clearly, we have clarified in the abstract and modified the last paragraph of the discussion section and overall, the conclusion section adding the relevance that this work brings to the field as a potential strategy for the treatment of ischemic cardiovascular diseases. Moreover, we have stated the further work that is necessary to achieve this goal.

  1. Figure 5 and Figure 6: The text on the axes must be enlarged. It is not clearly visible.

We appreciate this observation and agree with the reviewer these set of text requires bigger font size. Along with other reviewers’ observations, they will see that figures are now presented in color coding in order to facilitate the comparison of the data within the different groups of study.

  1. The grammatical and typo errors should be revised:

Change CoCl2 to CoCl2

Change RPM to rpm.

…temperature.  Coverslips is modified with … temperature. Coverslips …

Change H2S to H2S.

We apologise these errors were not addressed previously. We have now checked our manuscript and have also modified: CoCl2 to CoCl2, RPM to rpm, and H2S to H2S in the following sections: 2.6, 2.7 and discussion section.

The conclusion should be a summarized version of the total manuscript. It is better to summarize what the author has discussed in the manuscript briefly. 

We appreciate this insightful comment from the reviewer, and we indeed believe the conclusion section can be further improved by summarizing results from this study. We have modified the conclusion and included the main points and takeaway results from the study.

I recommend publication of this paper after minor revision.

We thank the reviewer for the observations and comments, and we appreciate the consideration for publication after the revisions suggested.

Reviewer 3 Report

In the manuscript by Aranguren et al, Sodium thiosulphate-loaded liposomes control hydrogen sulphide release and retain its biological properties in hypoxia like environment. In this manuscript, authors have utilized sodium thiosulphate as hydrogen sulphide donor, which is well known for its cardioprotective activity, in which sodium thiosulphate was encapsulated into novel nanocarrier (liposomes). Manuscript is well designed and executed in terms of its potential for the treatment of hypoxia related cardiovascular disease like ischemic disease. Manuscript could be improved by addressing following comments and suggestions before publication.

-Why did author choose Cocl2 to induce hypoxic-like environment, instead cells could be exposed low oxygen (hypoxia) chamber.

-Rationale of choosing DOTAP lipid for liposomes? DOTAP lipid are cationic in nature and cationic liposomes may attract more proteins in the systemic circulation and may hinder targeting or drug release.

-Degree Celsius (°C) writing format is not uniform, so please make it uniform.

-Supplementary Figure should go to Supplementary information file.

-Figure 2B, it’s better to present this image first as individual followed by merged image to show cellular structure and liposomal uptake.

-Figure 4C, regarding wound closure, its better to present wound closure (%) in actual term, it looks confusing because overall wound closure is 1%. Does it mean 100% closure? And Non-encapsulated STS looks higher wound closure than liposomal STS.

Author Response

Response to Reviewer 3 Comments

We respectfully submit revisions, following peer-review for our article manuscript entitled: “Sodium thiosulphate-loaded liposomes control hydrogen sulphide release and retain its biological properties in hypoxia-like environment”.

Below, we addressed comments detailing the revisions in the manuscript and our responses to the reviewers' comments are highlighted in blue font.

We thank the reviewer for the time taken to revise our manuscript and to suggest these very valuable insights to improve our manuscript.

-Why did author choose Cocl2 to induce hypoxic-like environment, instead cells could be exposed low oxygen (hypoxia) chamber.

We appreciate this observation. The rationale behind using CoCl2 as chemical inductor of hypoxia-like conditions, is that we aimed to use a model in which we would have stable expression of hypoxia inducible factors 1α under normoxic conditions (Munoz-Sanchez, 2019). Although we recognise the use of a chamber would be potentially more representative of the effects that low oxygen may have, it has been reported that short exposure to normal oxygen pressure, may significantly affect HIF-α expression, which we believe may lead to false or inconclusive results. For example, when isolating mRNA and or during assays such as real-time bioenergetics, unfortunately we are not in the capability in our laboratories to perform all these experiments and processes with continued hypoxia, leading to transient re-exposure to normal oxygen levels and possibly, altered HIF-1α expression. We believe that intermittent hypoxia may have not allowed us to observe the effects of our novel liposome encapsulating STS.

Muñoz-Sánchez J, Chánez-Cárdenas ME. The use of cobalt chloride as a chemical hypoxia model. J Appl Toxicol. 2019 Apr;39(4):556-570. doi: 10.1002/jat.3749. Epub 2018 Nov 28. PMID: 30484873.

-Rationale of choosing DOTAP lipid for liposomes? DOTAP lipid are cationic in nature and cationic liposomes may attract more proteins in the systemic circulation and may hinder targeting or drug release.

We appreciate this insightful comment from the reviewer and have clarified in text page 8, line 320 that ‘DOTAP is a positively charged lipid thus was selected to increase entrapment of the negatively charged thiosulphate ion’. DOTAP has indeed been used previously in the entrapment of negatively charged molecules such as RNA. We recognize further in vivo work would need to be done in order to assess systemic response to our formulation and have clarified the need for in vivo work in the discussion.

-Degree Celsius (°C) writing format is not uniform, so please make it uniform.

We appreciate this observation and have corrected this in line 156

-Supplementary Figure should go to Supplementary information file.

We thank the reviewer for this observation. The system requires for first submission to include all the information, including supplementary data within the main text. We will take this observation into account and request more clarification from the editors as the reviewer kindly observed.

-Figure 2B, it’s better to present this image first as individual followed by merged image to show cellular structure and liposomal uptake.

We thank the reviewer for this observation. We have now included individual images for DilC, DAPI and Merged to Figure 2.

-Figure 4C, regarding wound closure, its better to present wound closure (%) in actual term, it looks confusing because overall wound closure is 1%. Does it mean 100% closure? And Non-encapsulated STS looks higher wound closure than liposomal STS.

We appreciate this observation and agree with the reviewer the presentation of the scratch assay as grouped data is rather confusing. We have now modified this graph and presented the data as XY graph. The data is presenting the wounding area as a fold change from the initial time (0h) when the wound was created. Therefore, the wound area is reduced by time because of the migration of the cells and closure of the wound. We hope the improved presentation clarifies that liposomal STS displays higher wound closure when compared to non-encapsulated STS at 24h.

Round 2

Reviewer 1 Report

The reviewer suggests that the manuscript "Sodium thiosulphate-loaded liposomes control hydrogen sulphide release and retain its biological properties in hypoxia- like environment" looks better after the revision. However, it still lacks animal experimentation which is a major drawback of this study. However, considering the authors' inability to do animal experiments, the reviewer suggests that the manuscript can be published after addressing the following comment:

1. Delete "For the first time" from line no 18.